# Caspase-6 Is a Non-Apoptotic Effector of Shear-Induced Morphological Adaptation in Pulmonary Artery Endothelial Cells In Vitro

**DOI:** 10.3390/cells14211669

**Published:** 2025-10-25

**Authors:** Corey Wittig, Emir Bora Akmeriç, Laura Michalick, Jakob M. König, Wolfgang M. Kuebler, Holger Gerhardt, Robert Szulcek

**Affiliations:** 1Laboratory of In Vitro Modeling Systems of Pulmonary and Thrombotic Diseases, Institute of Physiology, Charité—Universitätsmedizin Berlin, Corporate Member of Freie Universität Berlin and Humboldt-Universität zu Berlin, 10117 Berlin, Germany; corey.wittig@charite.de (C.W.); jakob.koenig@charite.de (J.M.K.); 2Institute of Physiology, Charité-Universitätsmedizin Berlin, Corporate Member of Freie Universitätsmedizin Berlin and Humboldt-Universität zu Berlin, 10117 Berlin, Germany; laura.michalick@charite.de (L.M.); wolfgang.kuebler@charite.de (W.M.K.); 3Charité-Universitätsmedizin Berlin, Corporate Member of Freie Universitätsmedizin Berlin and Humboldt-Universität zu Berlin, 10117 Berlin, Germany; emirbora.akmeric@mdc-berlin.de (E.B.A.); holger.gerhardt@mdc-berlin.de (H.G.); 4DZL (German Centre for Lung Research), Partner Site Berlin, 10178 Berlin, Germany; 5DZHK (German Centre for Cardiovascular Research), Partner Site Berlin, 10785 Berlin, Germany; 6Integrative Vascular Biology, Max Delbrück Center for Molecular Medicine in the Helmholtz Association 13092 Berlin, Germany; 7Departments of Physiology and Surgery, University of Toronto, Toronto, ON M5S 1A1, Canada; 8Berlin Institute of Health, 10178 Berlin, Germany; 9Department of Cardiac Anesthesiology and Intensive Care Medicine, Deutsches Herzzentrum der Charité, 13353 Berlin, Germany

**Keywords:** caspases, caspase-6, shear stress, fluid flow, mechanoadaptation, mechanotransduction, morphological adaptation, endothelial cells, vascular biology

## Abstract

Caspases are known for their roles in cell death and inflammation. However, emerging evidence suggests they also mediate non-lethal processes, governed by a finely tuned balance of localization, activity, kinetics, and substrate availability. Given that many caspase substrates are implicated in mechanoadaptive processes, we investigated if caspases contribute to morphological adaptation of human pulmonary artery endothelial cells to fluid shear stress and other morphology-altering stimuli in vitro. Using selective inhibitors, we screened all major caspases for a role in endothelial cell adaptation to unidirectional laminar shear stress (15 dyn/cm^2^, 72 h). Selective inhibition of caspase-6, but not other caspases, impaired morphological shear adaptation. Only 5.5% of caspase-6-inhibited cells shear-adapted vs. 75.2% of vector controls. Live-cell FRET imaging revealed progressive caspase-6 activation starting at 18 h of shear stress, coinciding with the onset of morphological remodeling. The active caspase-6 localized predominantly perinuclearly, while caspase-3 remained inactive throughout shear exposure. Caspase-6 inhibition did not affect elongation in response to alternative biomechanical or biochemical stimuli, including uniaxial cyclic stretch (5%, 1 Hz), spatial confinement on narrow micropatterned RGD-lines, or TNF-α stimulation, nor did it impair cell adhesion, directed migration, wound healing, or barrier recovery after wounding. Our study uncovers a previously unidentified role of caspase-6 as a non-apoptotic, mechanosensitive effector specifically required for shear-induced morphological adaptation of pulmonary artery endothelial cells, highlighting a novel regulatory axis in vascular mechanoadaptation.

## 1. Introduction

Fluid shear stress is a fundamental biomechanical force generated by flowing blood, maintaining vascular homeostasis and governing endothelial functions [1]. In healthy vessels, unidirectional laminar flow promotes an anti-inflammatory, anti-thrombotic, and quiescent endothelial phenotype, whereas disturbed, non-linear, or oscillatory flow induces endothelial dysfunction, pro-inflammatory signaling, and maladaptive remodeling, processes implicated in atherosclerosis, pulmonary hypertension, and other vascular diseases [2,3]. Morphological adaptation of endothelial cells, particularly their elongation and axial alignment in the direction of flow-induced shear stress, is one of the most reliable indicators of functional shear responsiveness and serves as a structural hallmark of vascular health [4].

Shear forces are detected by specialized mechanosensors distributed across the cell surface [5]. At cell–cell junctions, PECAM-1, VE-cadherin, and VEGFR2 form a core mechanosensory complex that transduces shear into PI3K/Akt, Src, SMAD, and MAPK signaling [6]. At the basal surface, integrins and focal adhesion proteins transmit strain from the extracellular matrix to stress fibers and newly formed shear fibers [7]. At the apical surface, the glycocalyx and mechanosensitive ion channels such as Piezo1 regulate ion influx and cytoskeletal remodeling in response to flow [6]. These mechanosensory systems converge on common downstream effectors including Rho-family GTPases, flow-regulated transcription factors (e.g., KLF2, NRF2, NF-κB) and endothelial nitric oxide synthase (eNOS), which collectively orchestrate the structural and functional adaptation to shear stress [1].

Endothelial adaptation to shear stress occurs in distinct phases. Early responses, within seconds to minutes, involve rapid signaling through phosphorylation cascades, ion fluxes (e.g., Ca^2+^), and transient cytoskeletal changes [8]. Sustained adaptation over hours to days requires long-term structural adjustments including alignment and stabilization of actin filaments, formation of shear fibers, reinforcement of cell–cell junctions, reorganization of focal adhesions, and repositioning of the nucleus [7]. These late-phase changes not only establish the elongated and aligned morphology of endothelial cells, but also consolidate junctional and nuclear integrity. While the molecular basis of early signaling has been well characterized [7], mechanisms driving these late structural remodeling processes remain incompletely understood.

Caspases are a family of cysteine-aspartic proteases primarily recognized as initiators and executioners of apoptosis, as well as mediators of inflammation [9]. Based on function and sequence similarities, they are classified into initiator caspases (caspase-2, -8, -9, -10), executioner/effector caspases (caspase-3, -6, -7), and inflammatory caspases (caspase-1, -4, -5, -11) [10]. Caspases are synthesized as inactive pro-enzymes (zymogens) containing an N-terminal pro-domain, and large and small subunits [11]. Activation typically requires proteolytic cleavage, either via recruitment into activation complexes (e.g., the apoptosome or inflammasome) or by sequential cleavage cascades that release the active enzyme [12]. Once activated, caspases recognize short tetrapeptide motifs and cleave substrates after specific aspartate residues, thereby irreversibly modifying their targets [12]. Such substrate proteolysis can either inactivate the target protein or confer a gain-of-function by producing active fragments [13].

Beyond apoptosis and inflammation, increasing evidence indicates that caspases can function in non-lethal processes when their activation is spatially and temporally restricted [10]. Examples include neuronal pruning, myoblast differentiation, and stem cell regulation, where limited proteolysis of selected cytoskeletal or nuclear substrates enables remodeling without inducing cell death [14]. This kind of sublethal activity provides a mechanism for fine-tuning cellular architecture in response to developmental or environmental cues.

Caspase-6, encoded by *CASP6*, is classified as an effector mediating nuclear shrinkage and fragmentation during apoptosis, and possesses unique activation and regulation mechanisms distinct from other caspases [15]. Notably, caspase-6 is a relatively weak inducer of apoptosis compared to caspase-3 and caspase-7, often gets activated downstream of caspase-3 rather than being directly activated by initiator caspases, and can also undergo auto-activation under certain conditions [11,15]. Caspase-6 further differs from other caspases in its substrate specificity, preferentially cleaving proteins central to nuclear mechanics and cytoskeletal organization [16,17]. These caspase-6 substrates are particularly relevant to the late-phase structural remodeling required for sustained shear adaptation, such as nuclear repositioning and cytoskeletal realignment. Finally, caspase-6 has been uniquely implicated in neurodegenerative diseases such as Alzheimer’s and Huntington’s, where excessive caspase-6 activity and elevated levels correlate with axonal degeneration and neuropathological lesions [13,15]. Altogether, these properties make caspase-6 an intriguing candidate for regulating endothelial structural adaptation to mechanical stress.

Despite the recognition of non-apoptotic caspase functions, little is known about whether caspases contribute to endothelial mechanoadaptation. Prior studies of flow-induced endothelial remodeling have focused mostly on early sensing by kinases, ion channels, small GTPases, and transcriptional regulators [18,19,20], while potential roles for proteolytic remodeling in long-term shear adaptation have been largely overlooked. Hence, it remains unknown whether caspases participate in the structural remodeling phase that underlies endothelial elongation and alignment under sustained flow.

We hypothesized that caspases, through non-apoptotic proteolytic activity, contribute to shear-induced morphological adaptation of endothelial cells. To test this, we systematically inhibited individual caspases during exposure to laminar shear stress and monitored pulmonary artery endothelial cell (hPAEC) alignment and elongation. Our findings identify a novel role for caspase-6 in hPAEC shear adaptation, distinct from its traditional function in apoptosis.

## 2. Materials and Methods

### 2.1. Cell Culture

hPAEC were selected for experimentation due to their well-characterized temporal morphological shear responses [3], as well as their established shear-dependent roles in vascular homeostasis and disease [4,21]. Cells were obtained from PromoCell (#C-12241, Heidelberg, Germany) or Lonza (#CC-2530, Basel, Switzerland) and cultured on 0.1% gelatin-coated flasks in Endothelial Cell Medium supplemented with 5% FCS, 1% penicillin-streptomycin, and 1% Endothelial Growth Supplement (ScienCell, #1001, Carlsbad, CA, USA). Cells were maintained at 37 °C and 5% CO_2_, and culture medium was refreshed every other day. For all experiments, hPAEC were seeded at 40,000 cells/cm^2^ on fibronectin-coated surfaces (as RGD repeat peptides or 5 µg/mL human plasma fibronectin (Sigma-Aldrich, #F1056, Darmstadt, Germany), depending on assay), and used between passages four and six. At least three biological replicates were used per experiment.

### 2.2. Pharmacologic Inhibition of Caspases

Prior to any chemical or mechanical stimulation, hPAEC were pre-treated for one hour with 2 µM fluoromethyl ketone (FMK)-derivatized peptide inhibitors to allow cellular uptake, which act as irreversible and specific caspase inhibitors mimicking the effects of caspase deficiency without inducing cytotoxic effects [16]. Inhibitors were then maintained in culture medium throughout subsequent stimulation. Each inhibitor contains a tetrapeptide recognition sequence specific to its target caspases: Z-WEHD-FMK (caspase-1/4/5), Z-VDVAD-FMK (caspase-2), Z-DEVD-FMK (caspase-3/7), Z-VEID-FMK (caspase-6), Z-IETD-FMK (caspase-8/10), Z-LEHD-FMK (caspase-9), and Z-FA-FMK (negative FMK control) (Enzo Life Sciences, #ALX-850-239, Farmingdale, NY, USA). The full inhibitor panel was used in the initial shear stress screen to broadly assess caspase involvement. For subsequent experiments, only the caspase-6 inhibitor (Z-VEID-FMK) was used following the findings from the initial screen. To validate Z-VEID-FMK efficacy, caspase-6 activity was measured in hPAEC using FAM FLICA caspase-6 probes (Bio-Rad, #ICT095, Hercules, CA, USA), according to manufacturer’s instructions, following stimulation in triplicate with 200 nM staurosporine (Sigma-Aldrich, #569397, Darmstadt, Germany) for 4 h at 37 °C.

### 2.3. Unidirectional Fluid Shear Stress

Single-channel ibidi µ-Slides I Luer 0.4 (ibidi, #80176, Gräfelfing, Germany) were seeded with hPAEC and cells were allowed to adhere overnight. After pre-treatment with caspase inhibitors for one hour, hPAEC were subjected to 24 h of unidirectional, continuous laminar low shear stress (LSS, 2.5 dyn/cm^2^), followed by 72 h of unidirectional, continuous laminar high shear stress (HSS, 15 dyn/cm^2^) with the ibidi Pump System (ibidi, #10902, Gräfelfing, Germany). Five fields of view per channel were imaged every 24 h at 10× magnification (EVOS M5000, Invitrogen, Waltham, MA, USA), and cell orientation/elongation was quantified using a custom MATLAB R2022a (MathWorks, Natick, MA, USA) script, with at least 900 cells analyzed per condition. Cells were considered shear-adapted when aligned within 45° of the direction of flow and elongated (≥2× median static aspect ratio per donor).

### 2.4. Uniaxial Cyclic Stretch

UniFlex flexible-bottom membrane culture plates were seeded with hPAEC and cells were allowed to adhere overnight (Flexcell, #UF-4001P, Burlington, NC, USA). hPAEC were then either left untreated or pre-treated with caspase-6 inhibitors for one hour before uniaxial cyclic stretch was applied using the FX-6000 Tension System (Flexcell, #FX-6000T, Burlington, NC, USA) with the Arctangle Loading Station (Flexcell, #TT-4000-A, Burlington, NC, USA), delivering 5% strain at 1 Hz for 24 h. Six fields of view per well were subsequently imaged at 10× magnification (EVOS M5000, Invitrogen, Waltham, MA, USA), and cell orientation was assessed using Fourier components analysis of directionality with the Directionality 2.3.0 ImageJ plugin.

### 2.5. Micropatterned Spatial Confinement

hPAEC were suspended in culture medium with or without caspase-6 inhibitors, then seeded directly into ibidi µ-Slide VI 0.4 µ-Pattern RGD-coated slides patterned with narrow (20 µm) RGD-motif lines on a bioinert surface (ibidi, #83653, Gräfelfing, Germany). Cells were left to adhere and spread for 24 h. At least two fields of view per channel were subsequently imaged at 20× magnification (EVOS M5000, Invitrogen, Waltham, MA, USA). The aspect ratios of individual cells were manually measured in ImageJ 1.54p (National Institutes of Health, Bethesda, MD, USA), with at least 35 cells analyzed per condition.

### 2.6. TNF-α Stimulation

Twelve-well culture plates were seeded with hPAEC and cells were allowed to adhere overnight. The medium was then replaced with reduced-serum culture medium (1% FCS) for one hour, after which hPAEC were either left untreated or pre-treated with caspase-6 inhibitors for one hour. hPAEC were then stimulated with 10 ng/mL recombinant human TNF-α (PeproTech, #300-01A, Cranbury, NJ, USA) for 24 h. At least three fields of view per well were subsequently imaged at 10× magnification (EVOS M5000, Invitrogen, Waltham, MA, USA). The aspect ratios of individual cells were manually measured in ImageJ 1.54p (National Institutes of Health, Bethesda, MD, USA), with at least 30 cells analyzed per condition.

### 2.7. Electric Cell–Substrate Impedance Sensing (ECIS) Wounding

Impedance-based wound healing assays were conducted using the ECIS 1600R system (Applied BioPhysics, Troy, NY, USA) with 8W10E+ electrode arrays (Applied BioPhysics, Troy, NY, USA). hPAEC were seeded in duplicate wells, impedance readings were acquired every 18 s at 4000 Hz, and signal was allowed to stabilize (indicating adhesion, confluency, junctional maturation, and barrier stability). After stabilization, hPAEC were either left untreated or pre-treated with caspase-6 inhibitors. One hour later, electrical wounding was applied by delivering a high-frequency electrical pulse (60,000 Hz, 5 V, 20 s) to the cells through the electrodes to create 40 circular 250 µm diameter wounds per well. Electrode re-coverage with cells was monitored continuously through impedance readings to assess the dynamics of wound closure by cell migration and barrier recovery without the impact of proliferation.

### 2.8. Mechanical Scratch Wounding

Six-well cell culture plates were seeded with hPAEC and cells were allowed to adhere overnight. hPAEC were then either left untreated or pre-treated with caspase-6 inhibitors for one hour before scratch wounds were introduced manually using a 200 µL pipette tip. Wound closure was assessed at 0, 3, 6, 9, and 12 h post-wounding by imaging four separate fields of view along the wound at each timepoint. Wound width was measured at five randomly selected positions per image, and the rate of migration (ROM) was calculated at each timepoint in terms of wound width change per hour.

### 2.9. Confocal FRET Ratiometric Timelapse Imaging Under Shear Stress

To monitor caspase activity under fluid shear stress, hPAEC were transfected with a Förster resonance energy transfer (FRET)-based dual caspase-3 and caspase-6 biosensor (CFP-DEVD-YFP-VEID-RFP) linked in a single fusion protein, kindly provided and previously validated by Dr. Liusheng He [22]. The DEVD and VEID peptide sequences serve as specific cleavage sites for caspase-3 and caspase-6, respectively, where cleavage disrupts FRET (reduction in signal indicates active caspase). Transfections were performed using the HUVEC Avalanche Transfection Reagent (EZ Biosystems, #EZT-HUVE-1, College Park, MD, USA) following the manufacturer’s protocol. Following transfection, hPAEC were seeded into single-channel ibidi µ-Slides I Luer 0.4 (ibidi, #80176, Gräfelfing, Germany) and allowed to adhere overnight.

Imaging was performed using an LSM 980 confocal microscope (Zeiss, Oberkochen, Germany) with a Plan-Apochromat 20×/0.8 NA objective (Zeiss, #420650-9901-000, Oberkochen, Germany) in a 37 °C temperature-controlled chamber under atmospheric conditions. During acquisition, transfected hPAEC were subjected to a stepwise unidirectional, continuous laminar shear stress regimen: shear ramping of 5 dyn/cm^2^ and 10 dyn/cm^2^ for 30 min each, followed by 15 dyn/cm^2^ for 36 h using the ibidi Pump System (ibidi, #10902, Gräfelfing, Germany). hPAEC were maintained throughout in a custom CO_2_-independent endothelial basal medium (PromoCell, #C-22215, Heidelberg, Germany), modified to exclude phenol red and sodium bicarbonate, and supplemented with 1:1000 7.5% *w*/*v* sodium bicarbonate (Gibco, #25080094, Waltham, MA, USA) to maintain appropriate pH at atmospheric conditions, 1:1000 0.432% *w*/*v* β-glycerophosphate (Sigma-Aldrich, #G9422, Darmstadt, Germany), and EGM-2 BulletKit (Lonza, #CC-3162, Basel, Switzerland).

Donor, transfer, and acceptor fluorescence images were acquired every 7.5 min for a total of 36 h. Normalized FRET efficiency (NFRET) was calculated from flat-field- and background-corrected images as previously described [23].

### 2.10. Statistics

Statistical analyses and data visualization were performed in GraphPad Prism 10 (GraphPad Software, Boston, MA, USA), unless otherwise stated. Depending on the experimental design, either paired two-tailed Student’s *t*-tests or two-way ANOVA were used to assess significance, and Shapiro–Wilk tests were used to assess normality. When appropriate, Holm-Šídák multiple comparisons tests were applied following ANOVA. For repeated-measures ANOVA, the Geisser-Greenhouse correction was used to account for potential violations of sphericity.

Quantitative data is presented as mean ± SEM, unless otherwise specified. Statistical significance was defined as *p* ≤ 0.05, with *p*-values reported as follows: * *p* ≤ 0.05, ** *p* ≤ 0.01, *** *p* ≤ 0.001.

## 3. Results

### 3.1. Selective Inhibition of Caspase-6, but Not Other Caspases, Impairs Morphological Shear Adaptation of hPAEC

To investigate whether caspases contribute to the morphological adaptation of hPAEC to fluid flow, we performed a functional screen using specific caspase inhibitors during in vitro exposure of hPAEC to unidirectional, continuous laminar shear stress. The applied caspase inhibitors are cell-permeable, synthetic peptides that mimic the substrate of specific caspases and act by irreversibly binding to their catalytic site. Pharmacological inhibition was chosen over genetic suppression approaches because it enables direct assessment of caspase catalytic activity while excluding potential scaffolding functions of inactive caspases [24]. In addition, it avoids compensatory activation of other caspases known to occur with prolonged genetic knockdown of individual caspases [25]. hPAEC (n = 3 pooled donors) were shear-conditioned by applying a two-step regimen, 24 h of low shear stress (LSS, 2.5 dyn/cm^2^), followed by 72 h of high shear stress (HSS, 15 dyn/cm^2^). Caspase inhibitors were added one hour before shear initiation and maintained throughout the entire exposure period.

Under static and 24 h LSS conditions, hPAEC retained a typical unadapted morphology. In control cells, exposure to HSS induced progressive morphological adaptation, with cells becoming elongated and aligned along the direction of flow within 48 h. Similar morphological adaptation was observed in hPAEC treated with inhibitors of caspase-1/4/5 (WEHD cleavage site motif), caspase-2 (VDVAD), caspase-3/7 (DEVD), caspase-8/10 (IETD), and caspase-9 (LEHD). Strikingly, caspase-6-specific inhibition (VEID cleavage site motif) completely prevented the adaptive shear response, as hPAEC remained non-elongated and lacked directional alignment even after 72 h of HSS (Figure 1A). Z-VEID-FMK efficacy was validated in hPAEC using a FAM FLICA caspase-6 assay following staurosporine stimulation as an activator, confirming complete inhibition of caspase-6 activity (Appendix A).

Quantitative analysis confirmed the morphological differences. Shear adaptation was defined as dual fulfillment of cell alignment within 45° of flow plus an aspect ratio ≥2× the static median. While all groups appeared largely unadapted at early timepoints, morphological shear adaptation became increasingly prominent at 48 h and 72 h in all conditions except caspase-6 inhibition (Z-VEID-FMK) (Figure 1B). Specifically, vehicle control hPAEC increased from 15.8% at 24 h LSS to 75.2% shear-adapted cells following 72 h HSS, while caspase-6-inhibited hPAEC showed only minimal adaptation, changing from 1.2% at 24 h LSS to 5.5% after 72 h HSS. While potential off-target effects are a known limitation of FMK-based inhibitors, these were controlled for with the negative control, an identical FMK-derivatized peptide without the “VEID” caspase-6-specific sequence. In this control condition, shear adaptation proceeded normally compared to untreated cells.

Repeat experiments with four independent biological replicates confirmed the reproducibility of our findings. Compared to controls, caspase-6 inhibition significantly reduced shear adaptation at both 48 h (49.8% vs. 24.8%, *p* = 0.02) and 72 h (63.1% vs. 32.4%, *p* = 0.005) of HSS exposure (Figure 1C). This effect was not seen in a screen using endothelial cells from other vascular beds (Appendix A).

These results demonstrate that caspase-6 plays a non-redundant mechanoadaptive role in fluid flow-induced hPAEC remodeling, since inhibition of other caspases had no comparable effect.

### 3.2. Caspase-6 Is Not Required for Cell Elongation Induced by Non-Shear Stimuli in hPAEC

Having identified caspase-6 as specifically required for shear-induced morphological adaptation, we next investigated whether its role extends to other forms of hPAEC elongation not driven by flow. Hence, we evaluated the effects of caspase-6 inhibition under three well-characterized elongation stimuli in vitro: uniaxial cyclic stretch, micropatterned substrate confinement, and TNF-α stimulation.

First, hPAEC (n = 4) were cultured on flexible RGD-coated membranes and subjected to cyclic stretch (5% uniaxial strain at 1 Hz) for 24 h. Both control and caspase-6-inhibited cells displayed pronounced elongation perpendicular (90°) to the axis of strain, consistent with the known mechanobiological behavior of hPAEC (Figure 2A). Fourier components analysis of orientation distributions showed no significant difference between groups (*p* = 0.96; Figure 2B), suggesting that mechanosensitive elongation under stretch is unaffected by caspase-6.

Next, we assessed morphological responses to physical confinement using micropatterned substrates consisting of 20 µm wide RGD-motif lines. Both control and caspase-6-inhibited hPAEC (n = 3) adhered normally to the patterns and adapted their morphology to the underlying geometry, exhibiting elongation and alignment along the patterns (Figure 2C). Aspect ratio quantification confirmed no significant differences between groups (*p* = 0.82; Figure 2D), indicating that caspase-6 is not required for hPAEC elongation by spatial confinement.

Finally, we stimulated hPAEC (n = 3) for 24 h with 10 ng/mL TNF-α, a pro-inflammatory cytokine known to induce cytoskeletal remodeling and elongation in endothelial cells. Both control and caspase-6-inhibited cells responded to TNF-α with marked elongation (Figure 2E), with no significant differences in aspect ratio due to caspase-6 inhibition (*p* = 0.59; Figure 2F).

These findings demonstrate that caspase-6 is not broadly required for morphological adaptation of hPAEC in response to mechanical, environmental, or vasoactivating cues, but is specifically engaged during sustained flow-mediated mechanoadaptation.

### 3.3. Caspase-6 Inhibition Does Not Affect Directed Migration, Wound Healing, or Barrier Recovery After Wounding of hPAEC

Having established that caspase-6 is specifically required for morphological adaptation to shear stress, but not for elongation under other stimuli, we next examined whether caspase-6 activity contributes to hPAEC functional responses, particularly cell motility. Since both morphological adaptation and migration rely on actin cytoskeletal remodeling, we investigated whether caspase-6 influences directed migration and wound healing capacity in hPAEC.

To this end, we employed two complementary assays: electrical wounding with real-time impedance spectroscopy (ECIS) and mechanical scratch wounding. In ECIS-based migration assays, circular wounds (40 per well, 250 µm diameter each) were introduced by applying a high-frequency electrical pulse, and wound closure was monitored in real-time via impedance measurements as a proxy for hPAEC migration and monolayer reformation (n = 4). Post-wounding recovery dynamics were comparable between control and caspase-6-inhibited cells (*p* > 0.99; Figure 3A). Both groups restored baseline barrier resistance to similar levels (*p* = 0.83; Figure 3B), with a comparable total time to full recovery (~5 h, *p* = 0.44; Figure 3C).

Likewise, mechanical scratch assays in hPAEC (n = 3) revealed no significant differences in wound closure rate. At nine hours post-wounding, both groups showed similar levels of closure (Figure 3D), and quantitative analysis confirmed equivalent migration rates (*p* = 0.24; Figure 3E). In both conditions, complete closure was achieved in ~12 h.

These findings indicate that caspase-6 inhibition does not impair hPAEC motility, directed migration, or barrier recovery after wounding, further reinforcing that caspase-6 specifically regulates flow-induced remodeling.

### 3.4. Caspase-6, but Not Caspase-3, Is Specifically Activated After 18 H of High Shear Stress and Localizes Perinuclearly

To pinpoint the timing and subcellular localization of caspase activation during shear-induced adaptation, we performed live-cell confocal FRET imaging using a dual cleavage-sensitive fusion biosensor for caspase-6 (YFP-VEID-RFP) and caspase-3 (CFP-DEVD-YFP). hPAEC were subjected to HSS for 36 h, with ratiometric FRET imaging captured at 7.5 min intervals to monitor caspase activity in real-time.

Normalized FRET (NFRET) quantification revealed that caspase-3 activity remained largely unchanged across the time course, showing minimal deviation from baseline in all tracked cells (Figure 4A). In contrast, caspase-6 activity began to rise after approximately 18 h of HSS and increased progressively across all tracked cells. Caspase-6 activity significantly diverged from caspase-3 at 32.75 h (+10.3% from baseline, *p* = 0.05), reached maximal activation at 33 h (+11.9%, *p* < 0.001), and remained elevated through 36 h (+11.3%, *p* = 0.005).

Pseudocolored FRET ratio maps provide spatial insights into this dynamic. Caspase-3 remained inactive, while caspase-6 exhibited distinct perinuclear-localized activation from ~20 through 36 h (Figure 4B). This subcellular pattern was consistent across all tracked cells.

These findings establish that caspase-6, but not caspase-3, is specifically and temporally activated during sustained HSS. Activation coincided with the onset of visible morphological adaptation, and its perinuclear localization suggests that caspase-6 may function as a non-apoptotic effector linking mechanical force to perinuclear cytoskeletal and/or nuclear remodeling.

## 4. Discussion

Fluid shear stress is a fundamental biomechanical regulator of endothelial structure and function, promoting alignment, quiescence, and anti-inflammatory phenotypes in vascular homeostasis [1]. In this study, we identified caspase-6 as a non-apoptotic, shear-specific effector required for morphological adaptation of human pulmonary artery endothelial cells to sustained laminar fluid flow. Pharmacological inhibition of caspase-6, but not other caspases, markedly reduced elongation and alignment under shear stress. Interestingly, caspase-6 inhibition did not impair responses under cyclic stretch, spatial confinement, or TNF-α stimulation, nor did it affect adhesion, spreading, migration, wound healing, or barrier recovery. These findings indicate that caspase-6 has a specific role in mechanoadaptation of hPAEC to sustained laminar flow.

Live-cell FRET imaging revealed that caspase-6 activation begins only after ~18 h of high shear stress, coinciding roughly with the onset of visible morphological adaptation. Activity localizes predominantly to the perinuclear region. In contrast, caspase-3 remained inactive throughout, consistent with the specificity of caspase-6 in shear-adaptation. This delayed, spatially restricted activation pattern suggests that caspase-6 operates downstream of initial mechanosensing, potentially contributing to a late-stage remodeling phase that establishes and stabilizes the shear-aligned endothelial phenotype.

Caspases are best known for their role in cell apoptosis and as mediators of inflammation [9]. However, accumulating evidence now demonstrates spatially and temporally restricted non-lethal functions in diverse cell types [10,14], including barrier stabilization and regulation of membrane proteins [26,27]. Caspase-6 in particular has been shown to cleave cytoskeletal, cytoskeleton-associated/organizing/modulating, and nuclear structural proteins in human neurons in ways that allow remodeling without triggering cell death [17,28]. These findings support a model in which sub-lethal, substrate-specific proteolysis enables precise reorganization of structural components. Our findings extend this concept to vascular endothelial cells, revealing that caspase-6 activity is required for the morphological changes that underlie adaptation to shear stress. This is supported by the fact that shear mediated vascular remodeling is not driven by apoptosis [29].

A striking aspect of our data is the apparent specificity of caspase-6 for shear-induced morphological adaptation, but not other morphology-changing stimuli. While the precise upstream mechanism underlying the selective activation of caspase-6 by shear stress remains to be elucidated, this reflects key differences in how various mechanical and biochemical stimuli are sensed and transduced.

Uniaxial cyclic stretch and laminar shear both induce cell elongation but activate distinct pathways. Stretch applies bidirectional strain at the basal surface, sensed by focal adhesions and stretch-activated channels leading to Rho GTPase family-dependent reorientation of the cells and their actin stress fibers perpendicular to the stretch axis [30,31]. In contrast, laminar shear is sensed primarily at the luminal cell surface and requires cell–cell junctions [32]. This junctional signaling network comprises a mechanosensory complex encompassing PECAM-1, VE-cadherin, and VEGFR2, as well as flow-responsive receptors such as BMP, Notch, and Piezo1 [6,33]. This network activates SMAD, PI3K, and MEK/ERK aligning both the actin cytoskeleton and junctions along the flow axis.

Other elongation cues rely on different mechanisms. TNF-α induces elongation through TNF-receptor signaling activating NF-κB and MAPKs, with RhoA/ROCK orchestrating actin cytoskeletal remodeling and redistribution of adhesion molecules through phosphorylation cascades rather than proteolysis [34]. Collective cell migration and wound healing depend on cooperative cell polarization, transient and localized formation of actin-rich protrusions, focal adhesion turnover, and actomyosin based contraction, involving a spatiotemporally coordinated activation of the Rho GTPases RhoA, Rac1, and Cdc42 [35]. These membrane structures are rapidly turned over and do not appear to require caspase-mediated protein cleavage. Physical confinement on micropatterns, in turn, induces elongation through passive geometric guidance, aligning actin bundles along adhesive tracks without force-dependent remodeling [36]. The absence of caspase-6 involvement in any of these contexts underscores its selective requirement for the unique remodeling demands of laminar fluid flow.

Although our study does not identify substrates of caspase-6, several candidates emerge from prior work: (i) Caspase-6 is the only caspase known to cleave the core nucleoskeleton protein lamin A/C [37], a critical determinant of nuclear mechanics and mechanotransduction [38]. Controlled lamin cleavage could increase nuclear plasticity, facilitate reorganization of the perinuclear cytoskeleton, and enable nuclear reorientation under shear. (ii) Caspase-6 cleaves different isoforms of the cytoskeletal microfilaments (actins), microtubules, and intermediate filament proteins such as vimentin and desmin, as well as actin regulators like cofilin or drebrin [17,28]. Proteolysis of these components could permit targeted remodeling to accommodate long-term stress fiber alignment without compromising global cytoskeletal integrity. (iii) Junctional adhesion, adaptor, and scaffold proteins such as the known caspase-6 targets vinculin and FAK may require partial cleavage [28] to permit turnover, redistribution, or reinforcement of adhesion complexes, thereby fine-tuning junctional stability and signaling during the transition to a shear-adapted state. Given that the LINC (Linker of Nucleoskeleton and Cytoskeleton) complex transmits mechanical forces from adhesion proteins via the cytoskeleton into the nucleus, we speculate this might represent another potential target [39].

We previously showed that caspase-dependent cytosolic cleavage of the adherens junction protein PECAM-1 disrupts endothelial shear sensing and adaptation in pulmonary arterial hypertension [3]. Whether caspase-6 contributes to vascular disease, as it does to neurodegenerative disorders such as Alzheimer’s, Parkinson’s, and Huntington’s disease, where cleavage of cytoskeletal proteins contributes to axonal degeneration, remains to be determined [13]. In contrast to these neurodegenerative disorders, caspase-6 activity may actually be beneficial in endothelial cells, facilitating alignment and stability under physiological laminar flow. A critical question is whether dysregulated caspase-6 activity, either excessive, insufficient, mislocalized, or directed at pathologically altered substrates, contributes to vascular pathology. Disturbed flow profiles could alter the timing, magnitude, or spatial distribution of caspase-6 activation, thereby disrupting the fine balance between functional proteolysis and structural stability. Such imbalance may impair endothelial alignment, weaken barrier integrity, promote maladaptive remodeling, and apoptosis. These mechanisms could be relevant to a large range of cardiovascular disease where abnormal shear patterns and endothelial dysfunction coincide [1].

A key unresolved question is how caspase-6 becomes activated under shear stress, given that it can be triggered through multiple mechanisms, including auto-activation in a non-apoptotic context [40]. Our study focuses specifically on pulmonary artery endothelial cells and lacks direct substrate identification or in vivo validation. Preliminary screening suggested that caspase-6-dependent shear adaptation may be restricted to PAECs under our experimental conditions. As shear adaptation kinetics differ across endothelial subtypes [41], this observation highlights potential vascular bed-specific regulation of mechanoadaptive signaling that merits further investigation. To identify caspase-6 substrates during shear adaptation, proteomic profiling under flow (including quantitative proteomics, N-terminal labeling, or substrate-trapping mutants [12,42]), could be applied. Activity-based probes might provide orthogonal extension of our FRET spatiotemporal signals by directly labeling active caspase-6 and its associated protein complexes, enabling further investigation of activation dynamics relative to junctional, nuclear, or cytoskeletal remodeling. Given that our fluid flow model applied exclusively continuous, unidirectional laminar fluid flow, detailed studies of how shear-magnitude, duration, and disturbed/turbulent/oscillatory flow patterns modulate caspase-6 activity would clarify its place in the mechanotransduction hierarchy. Finally, in vivo knockout models would determine whether caspase-6 contributes to vascular homeostasis or pathology.

## 5. Conclusions

In conclusion, we identify caspase-6 as a non-apoptotic, shear-specific effector required for hPAEC morphological adaptation to laminar fluid flow. Its activation is temporally delayed relative to early mechanosensing, spatially restricted to the perinuclear region, and distinct from other caspases. Caspase-6 is dispensable for other morphological and functional adaptations, highlighting its context-specific role. These findings broaden the functional repertoire of caspases and introduce protease-mediated structural remodeling as a critical component of vascular mechanobiology.

## Figures and Tables

**Figure 1 cells-14-01669-f001:**
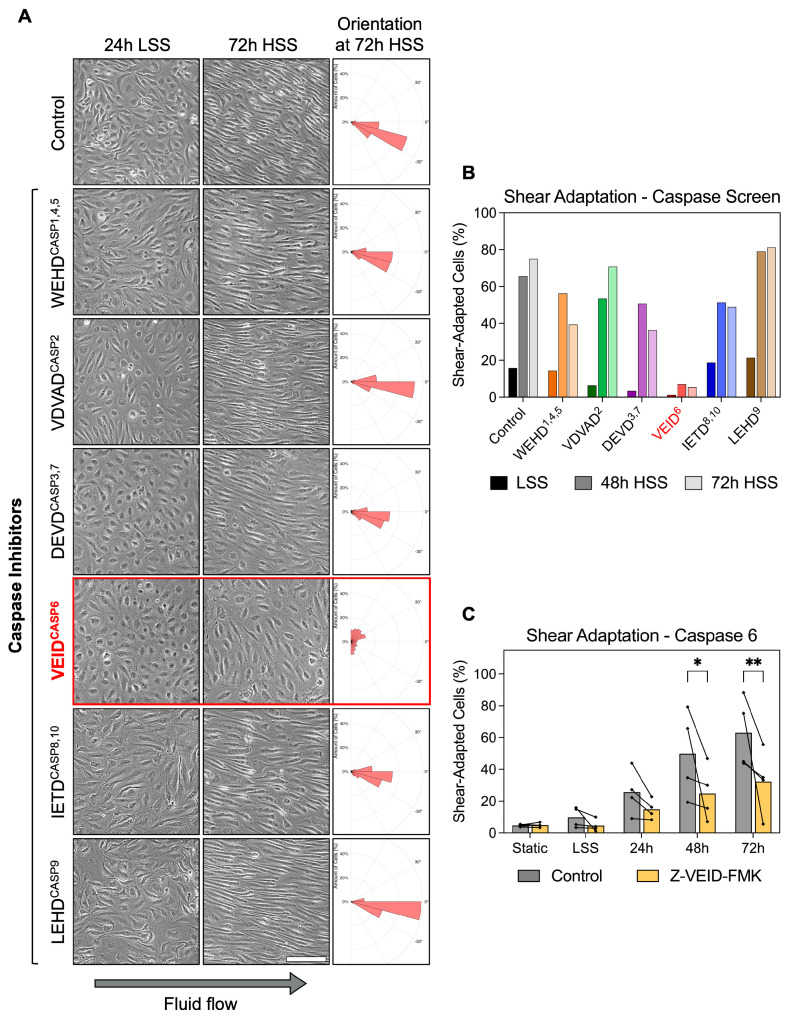
Caspase-6 is specifically required for morphological adaptation to fluid shear stress in hPAEC. (**A**) Representative images of hPAEC exposed to 24 h LSS (2.5 dyn/cm^2^) followed by 72 h HSS (15 dyn/cm^2^), with specific caspase inhibitors (cleavage site motif^inhibited caspase^, e.g., VEID^6^) or negative FMK control. Solely caspase-6 inhibition impaired the typical morphological elongation and alignment response of hPAEC to HSS. Orientation histograms generated using the *polarhistogram* MATLAB R2022a function (MathWorks). Scale bar = 200 µm. (**B**) Quantification of the inhibitor screen (n = 3 pooled donors). (**C**) Individual biological replicates (n = 4) confirmed significantly impaired morphological shear adaptation by caspase-6 inhibition (Z-VEID-FMK) after both 48 h (*p* = 0.02) and 72 h (*p* = 0.005) of HSS compared to uninhibited controls. *P*-values: * *p* ≤ 0.05, ** *p* ≤ 0.01.

**Figure 2 cells-14-01669-f002:**
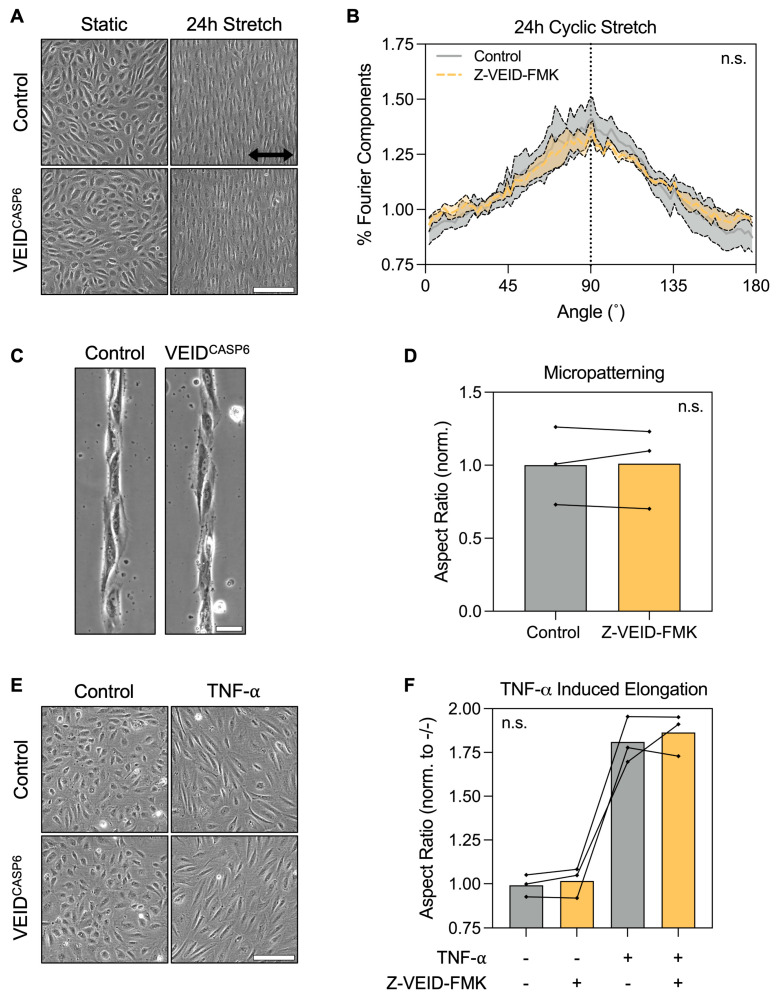
Caspase-6 is dispensable for non-shear morphological adaptation responses. (**A**) Representative images of hPAEC exposed to uniaxial cyclic stretch (5% strain, 1 Hz, 24 h) with or without caspase-6 inhibition (Z-VEID-FMK). Both groups elongated perpendicular to the axis of strain. Scale bar = 200 µm. (**B**) Fourier components analysis of directionality showed no significant difference in the directionality response to cyclic stretch with or without caspase-6 inhibition (*p* = 0.96, n = 4). (**C**) Micropatterned substrates (20 µm-wide RGD lines) induced hPAEC elongation along the pattern geometry irrespective of caspase-6 inhibition, with (**D**) no significant difference in aspect ratios (*p* = 0.82, n = 3). Scale bar = 50 µm. (**E**) TNF-α stimulation (10 ng/mL, 24 h) of hPAEC induced elongation irrespective of caspase-6 inhibition, with (**F**) no significant difference in aspect ratios between groups (*p* = 0.59, n = 3). Scale bar = 200 µm. “n.s.” = not significant.

**Figure 3 cells-14-01669-f003:**
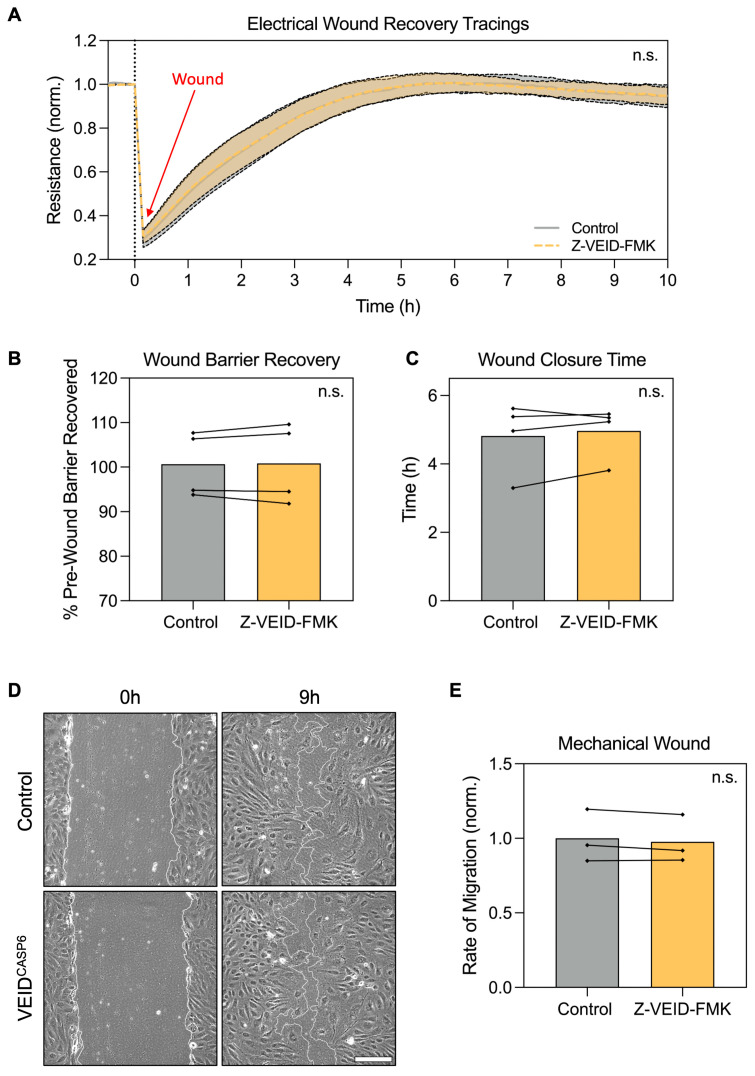
Caspase-6 inhibition does not affect hPAEC directed migration, wound closure, or barrier recovery after wounding. (**A**) Real-time ECIS migration assay following electrical wounding of hPAEC (n = 4) showed no difference in migration dynamics between caspase-6-inhibited cells (Z-VEID-FMK) and controls (*p* > 0.99). (**B**) Maximum barrier resistance recovered post-wounding and (**C**) total time to wound closure was unchanged by caspase-6 inhibition (*p* = 0.83 and *p* = 0.44, respectively). (**D**) Mechanical scratch assays showed similar wound closure dynamics irrespective of caspase-6 inhibition, with (**E**) no significant difference in migration rates (*p* = 0.24, n = 3). Scale bar = 200 µm. “n.s.” = not significant.

**Figure 4 cells-14-01669-f004:**
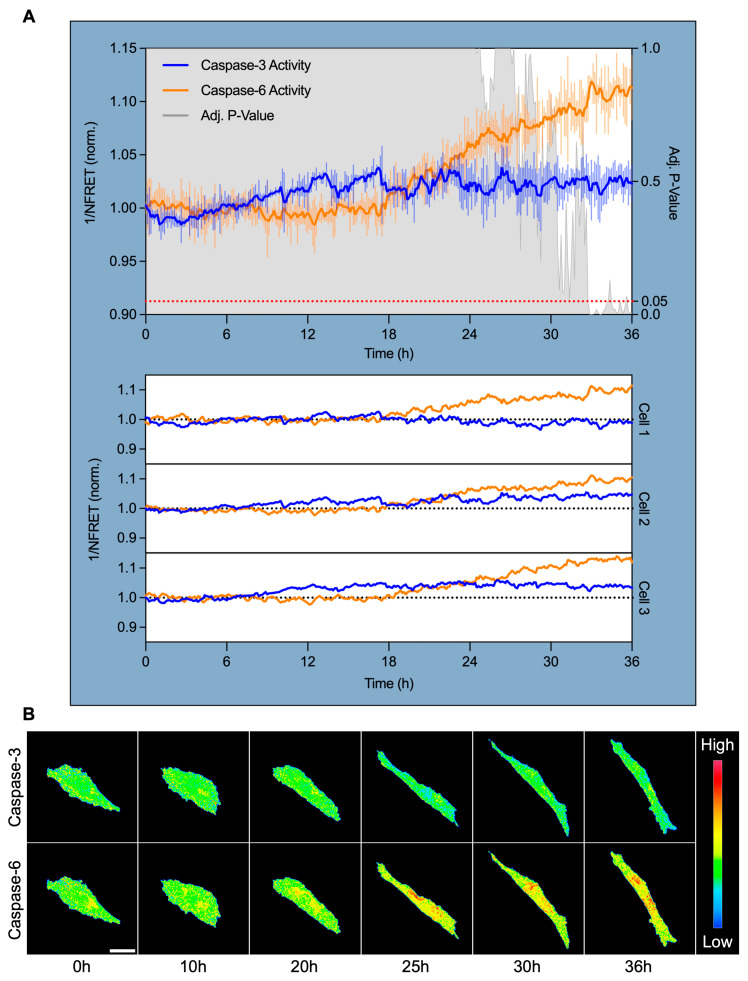
Caspase-6 activation coincides with morphological shear adaptation and targets the perinuclear region. (**A**) Live-cell FRET imaging of hPAEC expressing a dual caspase-6 and caspase-3 fusion biosensor revealed specific and sustained caspase-6 activation after 18 h HSS. Inverse normalized FRET (1/NFRET) values are shown, where increased inverse NFRET corresponds to increased biosensor cleavage/caspase activity. Caspase-3 remained non-activated throughout shear exposure, while caspase-6 activity began to increase after 18 h, diverged significantly from caspase-3 by 32.75 h (+10.3%, *p* = 0.05), peaked at 33 h (+11.9%, *p* < 0.001), and remained elevated at 36 h (+11.3%, *p* = 0.005). This was consistent across all tracked cells. The signal was smoothed using an exponential moving average with α = 0.2, while the error bars reflect raw, unsmoothed SEM. (**B**) Pseudocolored FRET ratio images show minimal caspase-3 activity, while caspase-6 activation localized predominantly to the perinuclear region. Scale bar = 30 µm.

## Data Availability

The raw data supporting the conclusions of this article will be made available by the authors on request.

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
