# Peer review of "Caspase-6 Is a Non-Apoptotic Effector of Shear-Induced Morphological Adaptation in Pulmonary Artery Endothelial Cells In Vitro"

_cells, 2025, doi:10.3390/cells14211669_

Round 1
Reviewer 1 Report
Comments and Suggestions for Authors
Witting et al. investigated the role of caspases in the morphological adaptation of human endothelial cells to fluid shear stress and other morphology-altering stimuli in vitro. This is a well-designed study that identifies caspase-6 as a contributor to stress-induced morphological changes. To further strengthen the conclusions, the following points should be addressed:
1. The study applied 24 h of continuous low laminar shear stress followed by 72 h of high laminar shear stress. What criteria were used to define the “low” and “high” shear stress levels? Do these magnitudes reflect physiological or pathophysiological conditions in vivo (e.g., venous vs. arterial ranges, disturbed vs. laminar flow)?
2. Is there a quantified shear stress–response relationship for the observed morphological changes, and can this response be attenuated in a dose-dependent manner by a caspase-6 inhibitor?
3. How does caspase-6 inhibition affect cytoskeletal architecture, particularly the distribution and integrity of lamin A/C? For instance, providing representative images and quantitative metrics (e.g., nuclear stiffness, lamin A/C intensity and organization, actin stress fiber alignment) under shear with and without caspase-6 inhibition would clarify the mechanistic link.
Reviewer 2 Report
Comments and Suggestions for Authors
The manuscript investigating the role of Caspase-6 as a Non-Apoptotic Effector of Shear-Induced
Morphological Adaptation in Pulmonary Artery Endothelial Cells is well written, and the conclusion is supported by the data included in the manuscript. The role of caspase-6 inhibition has been investigated; however, it will be interesting to know the effects of increased caspase-6 expression. Additionally, only drugs have been used to inhibit caspase-6; please support the data using siRNA or shRNA to attenuate its expression and the effects of shear stress. The wound healing assay images are included only for 9hrs (Figure 2A and E). Please include the images at 0 hrs and other time points. Further, what were the observations at 48 hours in Figure 1A.
Reviewer 3 Report
Comments and Suggestions for Authors
This is a well written report of well performed studies demonstrating the role of caspase-6 as a mechanosensitive effector.
Can the pharmacologic inhibitors used in this study elicit off target effects?
Can some of the major findings be confirmed using genetic approaches?
Does this mechanism shown in this study occur only in pulmonary artery endothelial cells?
What would be the clinical implications of the findings in pulmonary hypertension?
Reviewer 4 Report
Comments and Suggestions for Authors
This paper investigates the effects of fluid shear stress on morphological adaptation in endothelial cells. Specifically, the researchers revealed that caspase-6 is an essential non-apoptotic effector in the morphological changes induced by shear stress. This suggests that caspase-6 activation is involved in physical reorganization processes such as cell elongation and alignment, and is unrelated to cell elongation, cell migration, or wound healing induced by other physical or biochemical stimuli.
In this regard, I would like to make a few suggestions.
1. First, this paper notes that it did not directly identify the substrate of caspase-6, and the mechanism by which caspase-6 is activated under shear stress remains unclear. Therefore, I would like to know if there are any safeguards or alternatives for this.
2. This study was primarily conducted using human pulmonary artery endothelial cells (hPAECs). While hPAECs are a well-studied cell type for shear response, the limitations regarding whether the findings of this study can be equally applied to endothelial cells from other vascular sites (e.g., other arteries, veins, or capillaries) or endothelial cells from other species were not explicitly discussed. Shouldn't this cell type specificity be considered when interpreting the universality of the study results?
3. In this study, an inactive fluoromethyl ketone (FMK)-derived peptide inhibitor was used to suppress caspase-6 activity. While these pharmacological inhibitors are described as “mimicking the effects of caspase deficiency,” they carry potential limitations: they may exhibit off-target effects or provide incomplete or non-persistent inhibition compared to genetic approaches like gene knockout. The paper did not explicitly mention these common limitations of pharmacological approaches.
4. This study focused on unidirectional laminar shear stress conditions. However, in vivo blood flow can be significantly more complex, including disturbed, non-linear, or oscillatory flow. While the authors mentioned that disturbed flow patterns could regulate caspase-6 activity as a future research area, the limitation that the current study is restricted solely to these laminar shear stress conditions was not explicitly addressed regarding the physiological applicability of the present findings.
5. A key finding of this study is that caspase-6 is specifically activated by shear stress, yet is not required for other forms of morphological adaptation stimuli such as cyclic stretching, spatial confinement, or TNF-α stimulation. While the paper mentions that different stimuli activate distinct signaling pathways, it does not delve deeply into the precise upstream signaling mechanism by which caspase-6 is selectively activated by shear stress. Although the paper states that the caspase-6 activation mechanism is an “unresolved issue,” it does not explicitly mention the absence of an upstream mechanism explaining this specificity to shear stress as a limitation of the current study.
Reviewer 5 Report
Comments and Suggestions for Authors
This is a potentially interesting study describing the role of caspase-6 in morphological adaptation to shear stress, but not other kinds of stresses, such as exposure to TNFalpha or uniaxial stretch. The authors use peptides that irreversibly block the catalytic site of various caspases. Finally, the authors use FRET to show that caspase-6 indeed becomes active in the presence of shear stress.
This paper is potentially very interesting, but it lacks essential controls to confirm the results. This is the minimum that should be done:
- In this system, proof is needed that the inhibitors are actually inhibiting their caspase targets. This is particularly essential for caspase-6 and could be done by assessing the cleavage of a specific substrate in these cells, e.g. lamin A/C or FAK, which are involved in mechanotransducive pathways.
- Authors need to assess the efficiency of peptide uptake in these cells. This is conjoined with point 1.
- Authors need to confirm the effect of the inhibitor with a genetic approach. Since the authors use hPAEC, lentiviral shRNA delivery (e.g. Mission shRNA from Merck) would be the way to go.
- Authors need to perform the FRET experiment in cells treated with the caspase-6 inhibitor.
Round 2
Reviewer 2 Report
Comments and Suggestions for Authors
None
Author Response
Comment 1: None
Response 1: We thank the reviewer for their positive assessment and appreciate their time and effort in helping us improve our manuscript!
Reviewer 3 Report
Comments and Suggestions for Authors
Please incorporate all the responses in the revised manuscript.
Author Response
Comment 1: Please incorporate all the responses in the revised manuscript.
Response 1: We thank the reviewer for their appreciation of our responses, and have now incorporated the new data and our rebuttal round 1 responses directly into the manuscript. We have elected to not include the data on pulmonary arterial hypertension, however, as the paper is not focused on PAH and the PAH data is from a previously published dataset (PMID: 39897541) using pulmonary microvascular endothelial cells rather than PAECs.
Line-by-line changes (see tracked changes version for proper line numbers):
- “Can the pharmacological inhibitors used in this study elicit off target effects?”
- Now addressed on Lines 280-283.
- “Can some of the major findings be confirmed using genetic approaches?”
- Experimental rationale now addressed on Lines 253-257.
- New figure included that demonstrates efficacy of pharmacologic Z-VEID-FMK caspase-6 inhibition in our experimental model (Supplementary Figure S1).
- “Does this mechanism shown in this study occur only in pulmonary artery endothelial cells?”
- New figure included that tests caspase-6 inhibition with shear stress in three additional endothelial cell subtypes (hAoEC, hPMEC, HUVEC) now included (Supplementary Figure S2).
- Manuscript generally changed to specify the results were identified in pulmonary artery endothelial cells rather than more broadly endothelial cells.
- Cell type is now included as a limitation in the Discussion, incorporating the findings from Supplementary Figure S2 (Lines 497-502).
- “What would be the clinical implications of the findings in pulmonary hypertension?”
- As mentioned above, we elected not to include this data in the manuscript due to the cell type, the pre-published nature of the data set from which the data was derived, and the general aim of the manuscript.
In summary, we have now revised our manuscript in accordance with each of the reviewer’s points, and would like to stress our appreciation of their efforts and time, as we feel the reviewer’s input has significantly strengthened our manuscript.
Reviewer 5 Report
Comments and Suggestions for Authors
The authors have addressed only one point, which is the demonstration of the inhibition of caspase-6. The other points of my review were rebutted, but the arguments are not convincing. The point of asking for some experiments is to confirm the experimental observations, thus the argument that these experiments are not worth performing because the results are predictable is a weak one. This applies to peptide uptake (not addressed), The authors have not strengthened the paper sufficiently for this reviewer to change their position.
Author Response
Comment 1: The authors have addressed only one point, which is the demonstration of the inhibition of caspase-6. The other points of my review were rebutted, but the arguments are not convincing. The point of asking for some experiments is to confirm the experimental observations, thus the argument that these experiments are not worth performing because the results are predictable is a weak one. This applies to peptide uptake (not addressed), The authors have not strengthened the paper sufficiently for this reviewer to change their position.
Response 1: We respectfully but firmly disagree with the reviewer’s assessment. The revised manuscript includes additional validation data and clarifications that directly address the reviewer’s major concerns. We have also expanded the Methods and Supplementary Figures to provide additional experimental confirmation of our conclusions.
- Peptide uptake efficiency.
We believe the reviewer’s concern is misplaced in the context of this study. The question of peptide uptake is not critical to our conclusions because the inhibitor’s functional efficacy was directly validated. As shown in the newly added validation assay (Supplementary Figure S1), pre-incubation with 2 µM Z-VEID-FMK completely suppressed caspase-6 activation within 4h of stimulation with staurosporine, a potent caspase-6 activator. This confirms effective intracellular delivery during the 24 to 72h experimental window.
FMK-based caspase inhibitors are among the most widely used and well-validated reagents for probing caspase activity in living cells [PMID: 29563882, 15368275, 32366890, 12181741, 10381525, 36914635, amongst many others]. Our study therefore builds on established methodology rather than re-characterizing a compound whose uptake and pharmacodynamics have already been repeatedly documented in the literature. We have, however, now further stressed in the Methods that cells were pre-conditioned with the inhibitor before all stimulations (throughout Methods), and that ensuring cellular uptake was the purpose for pre-incubation (Lines 142-143).
- Genetic approaches (siRNA/shRNA).
We acknowledge the reviewer’s suggestion but emphasize that our goal was to assess stimulus-dependent enzymatic activity, not expression level. Partial gene silencing by siRNA/shRNA rarely achieves complete knockdown and is known to trigger compensatory up-regulation of other caspases [PMID: 16433925, 36674473, 11062535]. Because shear stress induces relatively weak, non-apoptotic activation of caspase-6, partial suppression of its expression would not be expected to replicate the complete functional inhibition achieved with FMK peptides. Moreover, genetic knockdown cannot directly assess activation kinetics under mechanical stimuli, while pharmacological inhibition specifically enables direct assessment of caspase catalytic activity while avoiding potential scaffolding functions of inactive caspases. Hence, genetic approaches are neither specific nor stable enough for our rather long experimental measurements.
Finally, FMK-based inhibitors remain the method of choice for probing caspase activity directly and are routinely accepted as sufficient mechanistic evidence in high-impact publications [PMID: 37390829, 28674081, 34802379]. We now explicitly explain this rationale in the Results (Lines 253-257).
- FRET assay under inhibitor conditions.
We appreciate the reviewer’s point; however, performing a FRET experiment under inhibitor treatment would be redundant and scientifically uninformative. The morphological shear assays already establish the complete functional loss of adaptation upon CASP6 inhibition, and the FRET assay was designed to provide temporal and spatial information on activation dynamics. We would like to make the reviewer aware that these long-term FRET experiments are extremely challenging to perform, wherefore we are not going to reproduce a result already demonstrated in multiple other experiments/replicates.
In summary, we maintain that our experimental design and the additional data provided fully support the conclusions drawn. The combination of validated inhibitor specificity, functional confirmation, assessment of numerous other forms of mechanical and chemical stimuli, and logical experimental rationale robustly demonstrates that caspase-6 is specifically required for endothelial morphological adaptation to shear stress in PAECs.
Round 3
Reviewer 3 Report
Comments and Suggestions for Authors.